# Toward the Definition of a Repertoire of Technical Professional Specialist Competencies for Operating Room Nurses: An Ethnographic Study

**DOI:** 10.3390/healthcare12171774

**Published:** 2024-09-05

**Authors:** Francesca Reato, Alessia Bresil, Chiara D’Angelo, Mara Gorli, Dhurata Ivziku, Marzia Lommi, Giulio Carcano

**Affiliations:** 1Department of Health and Social Professions, ASST dei Sette Laghi, 21100 Varese, Italy; francesca.reato@uninsubria.it; 2Educational Coordinator Master in Operating Room Nurses, University of Insubria, 21100 Varese, Italy; 3Operating Room Department, ASST dei Sette Laghi, 21100 Varese, Italy; alessia.bresil@asst-settelaghi.it; 4Department of Psychology, Università Cattolica del Sacro Cuore, 20123 Brescia, Italy; chiara.dangelo@unicatt.it (C.D.); mara.gorli@unicatt.it (M.G.); 5Department of Health Professions, Fondazione Policlinico Universitario Campus Bio-Medico, 00128 Rome, Italy; 6Department of Clinical and Molecular Medicine, Faculty of Medicine and Psychology, University La Sapienza, 00157 Rome, Italy; marzia.lommi@uniroma1.it; 7Department of Medicine and Innovation Technology (DiMIT), University of Insubria, 21100 Varese, Italy; giulio.carcano@uninsubria.it; 8Department of General, Emergency and Transplantation Surgery, ASST dei Sette Laghi, 21100 Varese, Italy

**Keywords:** competency, ethnography, nurses, operating room nursing, perioperative nursing, perianesthesiological nursing, repertoire

## Abstract

Registered nurses in the operating room require specialized competencies that surpass basic educational training. Existing national and international documents attempt to outline these competencies but often lack comprehensive details. To address this, a repertoire of technical and professional competencies for operating room nurses, aligned with European and National Qualifications Frameworks, is proposed. **Aim**: Develop a repertoire of technical and professional competencies for perioperative and perianesthesiological specialist nursing roles. **Methods**: An at-home ethnography design was employed, utilizing participant observation, interviews to the double, and focus groups. Convenience sampling included 46 participants from a university and a public hospital in northern Italy. Data were collected from September 2021 to June 2023 and analyzed using inductive content analysis and data triangulation. **Results**: Identified 17 specialized technical professional competencies for perioperative and perianesthesiological nursing, divided into 6 areas of activity. These competencies encompass 19 learning outcomes, 152 tasks, 222 knowledge elements, and 218 skills. **Conclusions**: This competency repertoire aids in the public recognition of qualifications and serves as a valuable tool for identifying, validating, and certifying competencies. Future research should focus on exploring the competencies of central sterilization nurses and transversal competencies.

## 1. Introduction

Perioperative nursing care consists of nursing assistance across preoperative, intraoperative, and postoperative patient care settings [1]. Perioperative clinical practice is very dynamic and in continuous evolution. Therefore, perioperative nurses need to continuously update their skills to fit with the developments in medical technology and evolving patient needs. This suggests that registered nurses working in this setting need specialist knowledge and skills that go beyond basic science education.

To acquire and practice specialized roles, nurses need to undergo specific post-general education programs accredited by governmental agencies and, where available, a formal system of credentialing linked to defined educational qualifications [2]. There is an international debate regarding the definition of roles, training, and competencies for specialist and advanced practice nurses and implementation within different clinical practices in European Nations [3]. Furthermore, a large diversity in nursing education exists at the different levels of the degree programs in nursing within and between nations [4]. This renders the identification and recognition of roles, competencies, and educational levels difficult, as well as international comparability, and underscores the importance of enhancing standards for education, certification, and regulation for specialist nurses [5].

Similarly to other specialist nurses, significant differences persist within each country for operating room nursing specialization [1]. Some education programs are exclusively tailored for specific roles such as operating room scrub nurses, circulating nurses, or anesthesia nurses. In contrast, others provide comprehensive training for all nursing roles within the operating room, including recovery room nurses and central sterilization nurses. In numerous countries, the specialization professional title is not a preferred or exclusive requirement for nursing practice in perioperative and perianesthesiological settings [1].

### 1.1. Challenges and Developments in Standardizing Competencies for Operating Room Nursing in Italy

In Italy, there is no unified approach to advancing the competencies of nurses in specialized fields such as operating room nursing. The operating room specialization is a post-basic education program, with a bachelor’s degree in nursing as the minimum entry requirement. The formal educational curriculum is designed to offer flexibility for different roles within the surgical setting. However, differences exist among the educational curricula based on regional areas and universities. Aligned with the Bologna Recommendations, the education requires a minimum of 60 ECTS credits (30 ECTS for theory and 30 ECTS for practice). Upon completion of the program, the university awards a certification that includes the operating room nursing specialization at Level 7 of the European Qualification Framework (EQF) [6]. In clinical practice, particularly in smaller operating rooms or settings requiring flexible resource allocation, maintaining a clear distinction between the roles of anesthesia nurses and scrub nurses is often impractical [7]. This results in interchangeable roles that include not only anesthesia and scrub nurses but also the operating room, recovery room, and central sterilization nurses. Additionally, specific professional qualifications are generally not required for these roles, with most nurses entering these positions relying on basic education and on-the-job training [7]. This trend toward multi-role assignments underscores the urgent need for specialized training and the definition of standardized competencies for each operating room nursing role to ensure effective and high-quality care across various Italian healthcare facilities [7,8].

Although the literature on operating room nursing offers a wealth of informative sources and manuals, in Italy, complete competency frameworks in bibliographic sources have not been found. In previous decades, various documents aimed to outline the activities and job descriptions of operating room nurses [9] or competency profiles [10,11]. These documents present a partial view of the competencies and exhibit confusion regarding the terms used to describe knowledge, competencies, and skills, which appear to overlap. More recently, the Italian Association of Operating Room Nurses (AICO) has collaborated with the National Federation of Orders of Nurses (FNOPI) to develop a competency framework for operating room nurses based on the European Operating Room Nurses Association (EORNA) proposed framework [1]. In a preliminary document [12], four areas of competencies were identified: professional practice/legal and ethical aspects, technical and professional areas, communication and relationship, and professional development and research. This framework primarily addresses the formal specialized education for operating room nurses, with competencies required for roles such as anesthesiologist nurse, scrub nurse, and circulating nurse indistinguishably overlapping.

### 1.2. Qualification and Competency Frameworks

To find more accurate and comprehensive documents on the competencies of operating room nurse specialization, one should consult the guidelines and standards issued by professional associations. The American Association of Operating Room Nurses (AORN) [13] published a guideline in 2021 regarding the scope and standards of practice for this specialization. This document defines 18 standards of practice, performance, and duties that all perioperative registered nurses, regardless of role, patient population, or specialty, are expected to competently perform. The EORNA proposed a framework for Perioperative Nurse Competencies in 2019, aiming to establish a standardized educational core curriculum for perioperative nurse specialization across Europe [1]. In this document, it is specified that perioperative nursing primarily encompasses care in surgical intervention, anesthesiology, and post-anesthetics. These diverse roles present challenges in competency development for perioperative nurses. These two documents primarily address the formal education required for specialization in operating room nursing. Despite these efforts, there is a need to “make transparent” the additional competency elements to better align with international standards. Specifically, it is crucial to identify the expected learning outcomes in terms of autonomy and responsibility, delineate the areas of activity associated with these competencies, and provide a detailed and accurate description of individual activities and tasks.

In recent decades, experts in education and governmental representatives from numerous nations across Europe have been working on the harmonization and standardization of qualifications in higher education. The European Qualification Framework (EQF) for lifelong learning, established in 2008 with the recommendation of the Council of the European Union 2008/C 111/01 [14] and updated in 2017 by the Council of the European Union Recommendation 2017/C 189/03 [15], became the reference framework for the translation of qualifications and levels of proficiency to improve the transparency, comparability, and portability of international qualifications. This recommendation suggested the use of the learning outcomes and competencies approaches, described in terms of knowledge, skills, autonomy, and responsibility, to nations. Following the EQF recommendations, the learning outcomes were articulated along three dimensions: knowledge (distinguished into theoretical and/or practical); skills (divided into cognitive and practical); and responsibility and autonomy (ability to apply knowledge and skills independently and responsibly). The qualifications framework facilitates the acknowledgment of skills not only between countries but also within the same country, across various learning contexts, including formal and vocational-lifelong and life-wide learning (non-formal and informal) settings [16].

Italy has been among the first nations to adhere to the Bologna Process and the European Community recommendations. Starting with Law 92/2012 and subsequent Legislative and Ministerial Decrees in 2013, 2015, and finally in 2018, it created the National Qualifications Framework and the National System for the Identification, Validation, and Certification (IVC) of Competencies. Italy has also produced important documents to facilitate the recognition and certification of qualifications and levels of proficiency. Among these documents, is found the Atlas of Work and Qualifications, which describes the national qualifications and the repertoire/repository of competencies for each qualification, including technical professional and soft skills [17]. In this Atlas, competencies developed by individuals in non-formal (life experiences) or informal (work) contexts, as well as competencies for Regulated Professions and Health Professions, are not present. This absence may be attributed to the mandatory evaluation and registration of these professions on official Professional Orders. Therefore, currently, in Italy, it is not possible to certify competencies acquired in non-formal (life experiences) or informal (work) contexts because they are not included in the national repositories. The same applies to technical professional nursing competencies. Nonetheless, this document provides thorough descriptions of the processes, methods, and tools for identifying, validating, and certifying competencies for other professional groups.

### 1.3. Research Gap and Study Aim

To the best of our knowledge, formal documents that present a complete repertoire of competencies and learning outcomes corresponding to knowledge, skills, activities/tasks, responsibility, and autonomy, aligned with EQF Standards [6] and the Italian National Qualification Framework Standards [17], are currently not available in Italy. Therefore, this research aims to develop a comprehensive repertoire of technical professional specialist competencies for operating room nurses in perioperative and perianesthesiological roles.

This study seeks to address the following inquiries:(a)Which competencies are demonstrated in perioperative and perianesthesiological operating room nurses?(b)Which specific activities and tasks, skills, and abilities for these two operating room nurses’ roles demonstrate the mobilization of the competencies?(c)What are the learning outcomes for operating room nurses’ competencies?(d)To which areas of activities do the competencies of the operating room nurses belong?

## 2. Materials and Methods

### 2.1. Design

This study used an ethnographic research design, specifically At-Home Ethnography. Conducting at-home ethnography entails using one’s familiarity with a particular environment to conduct research. It involves interpreting the behaviors, language, and materials observed within this setting, both from one’s own perspective and from that of other members of the organization [18]. This facilitates a more profound comprehension of real-life scenarios and captures a wider spectrum of activities.

### 2.2. Theoretical Framework

The development of the repertoire of technical professional specialist competencies for operating room nurses was guided by the European Qualifications Framework (EQF) [6] and the Italian National Qualification Framework [17]. Specifically, this repertoire includes (a) the titles of each competency, (b) the specific activities/tasks associated with each identified competency, (c) the knowledge and skills/abilities required for each competency, (d) the learning outcomes necessary for the development of each competency, and (e) the areas of activities to which the competencies pertain. Figure 1 illustrates these relationships.

These frameworks facilitated the structuring and systematic organization of competencies, activities/tasks, knowledge, and skills/abilities essential for the operating room nurses’ professional expertise. They provided a robust foundation for assessing and developing competencies within the specific context examined in the study.

In this study, as outlined in Legislative Decree No. 13 of 2013, competency is defined as “proven ability to use, in work, study or professional and personal development situations, a structured set of knowledge and skills acquired in formal, non-formal or informal learning contexts” [19].

Competencies are categorized into basic, technical professional, and transversal competencies. Technical competencies are defined as the “set of knowledge and skills related to the effective performance of specific professional activities in various fields/sectors” [20]. These competencies are crucial for performing specific tasks and responsibilities within a professional field, including the effective use of tools, equipment, and techniques [17]. In contrast, technical professional competency encompasses a broader range of abilities. It includes technical skills as well as the application of these skills within a professional context, incorporating communication, ethical decision making, critical thinking, and teamwork. This broader competency reflects a combination of technical expertise and interpersonal, cognitive, and behavioral attributes essential for effective professional practice. These competencies are fundamental to ensuring high standards of care and are central to the objectives of our research.

Knowledge refers to the result of assimilating information through learning, in a theoretical and/or practical way [6]. It encompasses the body of facts, principles, theories, and practices related to a specific field of work or study.

Skill refers to the capacity to apply knowledge and expertise to complete tasks and solve problems. Within the EQF [6], skills are categorized into cognitive and practical. Cognitive skills include the use of logical, intuitive, and creative thinking, while practical skills involve manual dexterity and the effective use of methods, materials, tools, and instruments.

Learning outcomes are the statements that describe what a learner knows and understands and is able to achieve upon the completion of a learning process [6]. They are articulated in terms of the learner’s acquired knowledge, skills, and level of responsibility and autonomy. Responsibility and autonomy refer to the learner’s capacity to apply their knowledge and skills independently and responsibly [6]. This involves making decisions, taking initiative, and managing tasks without direct supervision while ensuring accountability for their actions and outcomes.

This study focused solely on exploring the technical professional competencies of operating room nurses in perioperative and perianaesthesiological roles. Perioperative nursing encompasses all nursing interventions directly related to surgical procedures, including managing the operating field and instrumentation. In contrast, perianesthesiological nursing focuses on interventions related to anesthesia, covering the preparation room, operating room, and recovery room/post-anesthesia care unit. The substantial set of specific, coherent, and integrated activities/tasks, aimed at achieving a particular learning outcome and identifiable within a specific work process, are directed toward the corresponding areas of activity, as described in Figure 1.

### 2.3. Study Setting and Participants

Operating room nurse managers and nurses working at one public hospital in the northern region of Italy were invited to participate in the study. In addition, nurses enrolled in the Operating Room Master of specialization from one University in the North of Italy were involved. Data were collected from September 2021 to June 2023. The participation was voluntary and the sampling method was convenience-based. To be included, the operating room nurses and nurse managers were required to fulfill the following criteria: (a) to have worked in the operating room setting for at least 2 years, (b) to have experience of work in both roles, perioperative and perianesthesiological nursing, and (c) to have voluntarily and spontaneously accepted to participate in the study. The choice to include nurses with at least two years of experience was guided by Benner’s theory [21], which identifies this timeframe as sufficient for nurses to reach the “competent” level in specific roles. The nurses enrolled in the Operating Room Master of Specialization were included only if they met the following criteria: (a) they were actively registered in the program at the time of data collection and (b) they had completed at least 500 h of internship in the operating room. During these 500 h of clinical placements, students gained experience in scrub nursing, nurse anesthesia, and recovery room nursing, which was crucial for accurately identifying activities and tasks during participant observation.

We excluded participants (a) who were not proficient in Italian, (b) those whose primary experience was in the sterilization room, (c) and those who had been absent from work for an extended period (more than 6 months) or were out of work during the data collection period.

Before the research began, the healthcare institution was testing the interchangeability of nursing roles within the operating room and aimed to explore specific competencies associated with these different roles. Consequently, a joint research project was established with the university. The Directorate of Health Professions identified the operating rooms involved in this testing, and the operating room nurse managers were engaged in the research. These managers provided the contacts of nurses participating in the interchangeability program. Nurses who met the inclusion criteria and were actively involved in this program were invited to participate. Specialist Master’s nursing students conducted the data collection as a part of their internship activities.

The researchers presented the study to all invited participants, and those who were interested and motivated proactively contacted the researchers to participate. Overall, the sample included 6 nurse managers, 12 operating room nurses, and 28 nursing students registered in the Operating Room Master of specialization. Twenty-two nurses with less than two years of work experience and two students who did not complete the required 500 h of internship time were excluded. Throughout the study, there were no refusals or dropouts among the participants.

### 2.4. Data Collection

Data collection was conducted over six months in three distinct phases. Each phase aimed to comprehensively identify the maximum number of activities and skills associated with the two operating room nursing roles under investigation. The process continued until data saturation was achieved, at which point no additional activities or skills were identified, and data collection ceased.

Before data collection, participants received a four-hour training session on participant observation. This training included practical exercises in observing, field note-taking, and describing tasks and activities.

In the initial phase, direct participant observation was performed in the field, aligning with the principles of “at home ethnography”, which emphasizes participant observation as its central method [18]. The objective of this phase was to meticulously identify and describe each specific activity related to technical professional specialist competencies of operating room nurses in the two roles explored, achieving a high level of ethnographic depth. This approach focuses on the rigorous documentation and interpretation of observed social events without exploring the personal meanings or subjective experiences of participants. Participants observed five operating room nursing shifts per month, documenting all related perioperative and perianesthesiological nursing activities and taking field notes recorded in notebooks. The notes were brief to avoid disrupting the flow of observation but included essential details regarding who was doing what, when, and how [22]. Observers were instructed to record as many distinct activities as possible, each on a separate post-it note, ensuring no repetition. This method aimed to capture a comprehensive array of activities, contributing to a thorough description and facilitating the achievement of data saturation.

In phase two, 10 discussion group meetings were conducted with participants to systematically categorize each activity documented in phase one according to the specific roles and competencies of operating room nurses. Researchers facilitated brainstorming sessions to accurately associate these activities (see Appendix A for the open-ended questions addressed in these meetings). Each group meeting, which involved all participants, lasted two hours. Initially, overlapping activities—those with identical meanings but different descriptions—were either consolidated or eliminated. Exact duplicates were removed directly, while consolidation involved combining information on post-it notes. For activities listed under multiple competencies, group discussions determined the most appropriate competency, resulting in the activity being retained on the relevant board and removed from others. When an activity applicable to the perioperative phase was also relevant to the perianesthesiological phase, it was considered for inclusion in both, ensuring comprehensive coverage across both roles. Following this, the activities were chronologically organized according to their sequence in the operating room, from the beginning to the end of the shift. After the definition of the operating room nurse’s competencies for the two roles explored, the research team identified the learning outcomes for each competency, integrated the knowledge and skills required, and lastly associated the competencies to the areas of activities.

In phase three, the “interview to the double” [23], a data collection strategy consistent with the principles of “at home ethnography” was employed to further integrate data from previous phases. This method involves eliciting tacit routines—significant and recurring micro-actions enacted and interpreted by professionals—who articulate their daily practices, including attitudes and intentions behind their professional actions. Participants were asked to envision instructing a double who would replace them at work the next day, ensuring seamless continuity without detection. This unstructured in-depth approach included face-to-face interviews with specialist operating room nurses in the roles under study. Each interview focused on detailing the full spectrum of activities, knowledge, and abilities/skills required throughout a typical workday in the explored roles (see Appendix A for the open-ended questions addressed in the interview). The methodology facilitated a comprehensive narrative of procedural steps and activities, enriched by nuanced events and details, however minor. This data collection step integrated previous information regarding competencies in operating room nursing. All interviews were recorded and transcribed to facilitate subsequent data analysis, resulting in a total of 34 individual interviews conducted.

### 2.5. Data Analysis

Inductive content analysis and data triangulation were employed consistently across all phases of the research. The collected data, including interview transcripts, field notes from participant observation, documentation on post-it notes, and outcomes from group discussions, underwent a four-step analysis process: labeling tasks descriptively, categorizing patterns, outliers identification, and synthesizing within the framework [24]. Activities/tasks and knowledge/skills/abilities were coded and labeled within competencies; patterns were identified concerning tasks/activities, knowledge, and skills/abilities within specified competencies; outliers were flagged during the association of tasks/activities; and generalization and grouping of competencies were structured according to the competency framework (see Figure 1).

Throughout this process, the principles of rigor and reflexivity were applied in an integrated manner to ensure the validity and reliability of the findings [25]. Validation was ensured through participant discussions and regular team meetings. Reflections from group work meetings were compared and triangulated visually through graphical schemes accessible to all team members. Network diagrams were generated to depict interconnections among activities/tasks, knowledge, skills/abilities, learning outcomes associated with competencies, and professional activity areas. This approach facilitated the creation of a comprehensive map that enhanced visibility and usability for both the research group and participants. The network representation emphasized connections among diverse elements, fostering productive discussions on optimal modifications. The data that emerged from the entire analysis process were synthesized and systematized within the repertoire of competencies.

### 2.6. Ethical Considerations

The study complies with the ethical standards and the principles of the Declaration of Helsinki [26]. The Board of Directors of the hospital and the University authorized the research. The participants voluntarily adhered to the study and, after adequate information, signed the informed consent. Data access was restricted solely to the research team.

## 3. Results

Overall, 46 participants were enrolled in the study: 6 nurse managers, 12 nurses, and 28 nurses registered in the Operating Room Masters of specialization. The sample demographic data are presented in Table 1.

To establish the repertoire of technical professional specialist competencies for perioperative and perianesthesiological operating room nurses, a systematic multi-step approach was employed. Initially, observational data were collected through field participatory methods, categorizing them into specific activities and tasks associated with various competencies. Each participant conducted 30 observations in the operating room, resulting in a total of 1380 observations. From these, 393 distinct tasks/activities related to perioperative and perianesthesiological nursing were identified. These activities/tasks were then described, analyzed, and discussed during 10 team meetings. During these sessions, participants presented the identified tasks/activities recorded on post-it notes, eliminated duplicates, and aggregated similar tasks under common labels. They also proposed associations between these activities/tasks and specific competencies in operating room nursing. Multiple boards, each representing a competency, were used to map the identified tasks to the relevant competencies. Of all the activities/tasks identified, 152 activities/tasks were maintained and organized under 17 competencies. Refer to Table 2 for a summary of information on the results. For the full list of tasks/activities associated with the competencies, refer to Appendix A.

Technical professional specialist competencies were grouped under two roles for nurses in the operating room: perioperative nursing and perianesthesiological nursing. Seven technical professional specialist competencies related to perioperative nursing were identified: preparing the perioperative environment, restoring the perioperative environment, preparing the assisted person for the surgical procedure, managing perioperative nursing care, promoting perioperative care continuity, documenting planned and delivered nursing care in the perioperative setting, and promoting perioperative safety and risk management. Similarly, seven technical professional specialist competencies related to perianesthesiological nursing were identified: preparing the perianesthesiological environment, restoring the perianaesthesiological environment, preparing the assisted person for anaesthesiological conduct, managing perianaesthesiological nursing care, promoting perianaesthesiological care continuity, documenting planned and delivered nursing care in the perianaesthesiological setting, and promoting perianaesthesiological safety and risk management. Additionally, three competencies were common to both roles and could be performed interchangeably by operating room nurses: admitting the assisted person in the perioperative and perianaesthesiological environment, transferring and positioning the assisted person in the perioperative and perianaesthesiological environment, and promoting professional development, competencies enhancement, and best practices in perioperative and perianaesthesiological settings (Table 3).

After identifying competencies and associated tasks/activities, researchers conducted multiple “interviews to the double” to uncover additional tasks/activities not observed during participatory observation. Data saturation was reached after 34 interviews, each lasting 120 min. These interviews provided critical information for identifying the knowledge and skills/abilities required to perform the competencies relevant to the operating room nurses in the studied roles. In total, 222 knowledge areas and 218 skills/abilities were identified and associated with the 17 competencies described (see Table 2).

The inductive content analysis of the “interview to the double” generated 108 codes, 16 subthemes, and 4 themes (see Appendix A for details).

The first theme “Nursing care in routine and emergency settings” includes 2 subthemes and 16 codes that describe nursing activities performed routinely or in emergencies in perianestesiological and perioperative contexts. In these subthemes, operating room nurses described activities related to access to the operating room, the utilization of safety checklists, execution of specific procedures of the two roles, positioning of the operating field and the surgical team, verification of safety checks, management of biological material, and health education. Among the activities participants call to attention, for example, was the staff access to the operating room:

“*You have to access the perioperative setting by respecting the clean and dirty routes, this means that you will have to go through the locker room first which is a kind of filter zone, then you will have to wear the appropriate footwear, mask and headset and then enter the operating block. Once you arrive at the operating block, you will have to go to the corridor where the specific operating rooms are marked and identify the one in which your name is marked. You will then check which surgical specialty you are assigned to, and once you see which surgical specialty you have been assigned to, you will need to check the operating list for the day of that surgical specialty and from the operating list you will identify all the information useful for anesthesiological conduct*.” [participant 5, nurse, female, 28 years]

Another activity considered critical in the perianesthesiology setting is the “Application of the safety checklist”.

“*It is most important the reported list, the order in which the attendings are reported and therefore the type of anesthesia you will need to manage for each surgery, the type of monitoring and the medications you will need to use*.” [participant 3, nurse, male, 32 years]

The second theme “Operating room environment and resources” includes 2 subthemes and 28 codes that describe the procedures in the perianaesthesiological and perioperative environment. The nurses provided details related to tasks and activities performed to ensure the availability of material resources and the completion of safety checks necessary for the operation, as well as the verification of the environment and materials for the surgery setup. For example, the participants highlighted the “Setup of medical devices and electro-medical equipment”.

“*I normally […] head to the referring operating room and go and check that all the medical equipment and devices are there. I suggest you check not only that they are there but obviously that they are intact and working […] that there is the electroscope working in both the cutting function and the clotting function, that it has the bipolar socket and that this socket is compatible with our cables or that there is possibly the adapter […] that everything is turnable and working. I suggest you check that the scialitica (light) is working, that there are no burned out bulbs, and that the scialitica (light) can move to different positions*.” [participant 2, nurse, female, 36 years]

The third theme “Patient care in the perioperative and perianaesthesiological environment” includes 6 subthemes and 30 codes that describe the activities involved in patient care, from admission, preparation, and positioning to care and transfer. For example, the participants emphasized the importance of verifying the patient’s preparedness for surgical intervention using checklists.

“*We have a checklist for the preparation of the person assisted and there I go and see that everything that has been required for that surgery has been done and so I talk about trichotomy, removal of removable prostheses, jewelry and I also usually check that preoperative hygiene has been done, where the person has not done it (or colleagues have not done it) I proceed directly. I also go and check the marking of the surgical site. Obviously if it is required in organs that are not single such as the lower or upper extremities. Finally I obviously talk to the person being assisted, already hinting at what’s going to happen and then I proceed to health education, pointing out for example that they might be discharged from the operating room with a bladder catheter or one or two drains. [...] And finally, once I have done all this I perform antisepsis of the surgical site, place the sterile drapes over the patient, the clear membrane, and the actual surgery begins*.” [participant 8, nurse, female, 26 years]

Finally, the fourth theme “Promotion of Nursing Care Continuity and Clinical Governance” includes 6 subthemes and 34 codes that describe the roles the activities performed to ensure continuity of care, patient safety, and risk management through care planning and documentation.

“*There are some differences between the Anesthesia Nurse and the Scrub Nurse […] what differentiates them concerns specific nursing care planning, because the Scrub Nurse deals with the perioperative pathway and the Anesthesia Nurse with the perianesthesiological pathway. […] at the time of handover, the Scrub Nurse forwards all data and information regarding surgical procedures; the Anesthesia Nurse forwards all data and information regarding anesthesiological procedures. Also in terms of filling out the continuity of care form, there are differences for the perioperative specific one, where the Scrub Nurse describes everything regarding instrumentation and for the perianesthesiological specific one where the Anesthesia Nurse describes everything regarding anesthesiological conduct*.” [participant 7, nurse, male, 27 years]

“*In the documentation of nursing care in the perioperative setting, after making sure that the health documentation is present and complete, filling out the relevant part of the checklist […] I usually document the hours of surgery, so the beginning and the end of my surgery. The type of surgery, the number of drains if we put them in and we then go on to also document the type of dressing that we did and finally in the form we mark the final count of the gauze, tablets, needles and blades that were used that all have to go back. For the documentation competency of perianesthesiological nursing, after verifying that health care documentation is present and complete, filling out the relevant part of the checklist, I usually proceed by documenting all the monitoring that has been done, all the medications that have been used, and all the medical devices that have been placed*…” [participant 2, nurse, female, 36 years]

After thoroughly analyzing the tasks and competencies identified across the three phases of the study, the research team deliberated on the associated learning outcomes and grouped the competencies into specific areas of activity. Following extensive discussions and brainstorming, 6 activity areas were defined for the two operating room roles, and 19 learning outcomes were established. Complete details are available in Table 2 and Appendix A.

It is impractical to present the entire repertoire within this manuscript; therefore, the authors have chosen to showcase the complete repertoire for a single activity area as an illustrative example, presented in Table 4. For the full repertoire of competencies, please contact the first author of this manuscript.

## 4. Discussion

This study aimed to define and generate a comprehensive repertoire of technical professional specialist competencies for operating room nurses in perioperative and perianesthesiological roles.

A repertoire of competencies encompasses structured representations of professional content, serving as mandatory benchmarks for the identification, validation, and certification of competencies [27]. It acts as a critical informational resource for career guidance and labor market alignment, facilitating the matching of supply and demand. The repertoire can include both technical professional competencies and training path standards, which are essential for regulating educational trajectories and formulating active labor market policy measures [27].

The repertoire of competencies developed in this study can be applied in clinical practice according to national regulations and educational standards. In the following paragraphs, we will discuss the findings of this research in relation to the national and international literature and official documents. For improved comprehension, the discussion is divided into subtopics.

### 4.1. Specialized Operating Room Nursing in Italy: Regulatory Gaps and the Need for Targeted Education

Professional practice is regulated by national standards. In Italy, the regulation of the nursing profession is governed by the National Federation of Orders of Nursing Professions (FNOPI), which establishes the framework and scope of practice for nursing. Key legislative provisions, such as Law 251/2000, Law 42/1999, and Law 739/1994, granted nurses broad professional autonomy to perform a variety of tasks and activities commensurate with their training and competencies. Despite these provisions, current laws recognize general nursing qualifications but do not specifically address specialized areas, including operating room nursing. Additionally, there are no formal work contracts or detailed job descriptions within healthcare institutions that explicitly acknowledge specialized roles or detailed competencies for operating room nursing.

In previous decades, specialization in operating room nursing in Italy was distinctly defined. For example, in 1890, specialized operating room nurses had clearly defined responsibilities and skills, including meticulous cleanliness, disinfection of instruments, and precise preparation of surgical materials [9]. Regional Decree 225/74 also outlined specific tasks for anesthesia nurses. However, by the 1980s, the focus shifted toward a multifunctional nursing role aligned with the “unified nurse” philosophy. This shift led to a decline in specialized training in nursing, reducing the emphasis on operating room specialization [28]. Moreover, contemporary organizational models often require nurses to perform multiple roles within the operating room. The challenges and competency gaps observed among nurses underscore the urgent need for specialized training to develop the necessary skills. It is essential to align specialized education with work organization to prepare highly skilled professionals capable of delivering quality patient care. Operating room work organizations must also leverage all available flexibility mechanisms to fully utilize the knowledge, skills, and abilities of nursing professionals, rather than simply reallocating them to meet organizational needs [8]. Currently, while the legal framework supports autonomous nursing practice, it lacks specificity in recognizing and detailing specialized roles within operating room nursing, highlighting the need for renewed emphasis on specialized training and recognition.

Furthermore, it is crucial to establish clear guidelines regarding the educational requirements for competency training in operating room nursing. In Italy, undergraduate nursing programs are designed to prepare generalist nurses with foundational knowledge and skills applicable across a range of healthcare settings. Although some training in areas like sterile field management, anesthesia medications, and post-surgery care is provided, it is impractical to cover the full range of specialized skills required for operating room nursing within a bachelor’s degree. Specialized competencies, such as those needed for perioperative and perianesthesiological nursing, are typically acquired through advanced post-basic education following the completion of a general nursing bachelor’s degree as specified by EQF [6]. Nevertheless, currently, operating room nurses enter their roles with generalist training and receive additional on-the-job training, as formal specialization is not required for employment. This raises important questions about the competencies that should be emphasized in undergraduate and postgraduate educational programs. To ensure safe and high-quality care, it is crucial to legally recognize and support specialized training and post-basic education for nurses, as relying solely on generalist nurses may not adequately address the complexities of this demanding field. Therefore, the repertoire of technical professional competencies developed in this research can support the education of specialized operating room nursing roles and help evaluate the outcomes of on-the-job training and competency evolution.

### 4.2. Expanding Competency Frameworks in Operating Room Nursing: Challenges, Gaps, and a Comprehensive Approach

There has been a lack of consensus regarding the components of competency in the operating room nursing, as in this setting nursing competency is a complex multifaceted concept that is challenging to define and measure [29]. The terms “competencies”, “knowledge”, “skills/abilities”, “activities/tasks”, and “learning outcomes” are not always consistently defined or used with the same meaning in the official documents of nursing associations. Sometimes, only one or two of these concepts are addressed, while the others are omitted. For example, the AORN document [13] specifically refers to “Competencies” and “Standards” in a precise manner, deliberately omitting references to knowledge, skills/abilities, activities/tasks, and learning outcomes. The authors acknowledge that the document is not exhaustive and invite others to contribute to and expand upon their work. The EORNA document [1] clearly outlines the essential knowledge required for developing competencies. However, it does not explicitly describe the skills or abilities, which are likely embedded within the Performance Criteria and Key Indicators. In the document on Nursing Clinical Practitioner Standards in Pacific Island Countries and Territories [30], the authors chose not to use the terms skills, knowledge, abilities, or learning outcomes. Instead, the standards they describe closely resemble the activities listed in this study repertoire.

The technical professional competency repertoire developed in this study provides support and extension of the core competency frameworks established by operating room nursing associations such as EORNA [1], AORN [13], and AICO [12]. Unlike these frameworks, which predominantly address the general educational curriculum for perioperative nursing roles, this research focuses specifically on two distinct roles within the operating room. The repertoire elaborates comprehensively on the requisite knowledge and skills for core technical professional competencies, incorporating detailed performance criteria, task-specific knowledge and abilities, and activity areas pertinent to these roles. The distinction between perianesthesiological and perioperative nursing highlights the unique competencies required by each specialization. This enhancement complements the AORN standards [13] by providing granular descriptions of practice, performance, and duties specific to the two roles under investigation, thereby extending areas of autonomy and responsibility. The repertoire also adds substantial and in-depth information to the AICO profile [12]. Drawing from the two theoretical frameworks [6,17], this study integrates the core competencies of operating room nurses acquired through formal education with those developed through non-formal (life experiences) and informal (work) contexts. This integration facilitates the development of a specialized curriculum tailored to meet the clinical and theoretical demands encountered in operating room settings. This study provides a comprehensive description of all the elements that constitute competency in operating room nursing for the roles explored. It also clarifies the concept of competency, which is frequently used in nursing internationally but defined in various ways by different professional organs.

### 4.3. Gaps in Research and International Perspectives on Competency in Operating Room Nursing: A Comparative Analysis

It is important to recognize the limited amount of peer-reviewed research on competency definition in operating room nursing. Existing studies predominantly focus on developing instruments that address only certain aspects of competency, without thoroughly exploring the tasks, activities, or the comprehensive knowledge and skills required to fully utilize these competencies. Furthermore, there are few studies that examine the education, competency, and roles of nurses in specialized operating room roles from an international perspective. These studies often highlight significant cross-country differences in education duration and training content, job descriptions, responsibilities, and tasks [31]. This suggests that organizational differences between countries cannot be fully explained by educational factors alone. More nursing research is needed to enhance the literature on this topic exploring educational and organizational factors associated with competency mobilization in such contexts [32]. Consequently, despite conducting an extensive literature review, limited opportunities for comparison of this study’s findings with scientific research are evidenced. Instead, the authors have aligned the findings with the guidelines and official documents of operating room nursing associations.

A substantial body of the literature is available on the roles of nurse anesthetists. Organizations such as the International Federation of Nurse Anesthetists [33], the American Association of Nurse Anesthetists [34], and the International Council of Nurses [35] have outlined educational requirements and practice standards for advanced competencies of nurse anesthetists. Differences in practice scope have emerged between low- and high-income countries. In low-income countries, nurse anesthetists often act as the sole providers of anesthesia for surgical procedures, performing tasks such as preanesthetic evaluation, ordering preanesthetic medications, general anesthetic induction, tracheal intubation or extubation, spinal and epidural blocks, intraoperative anesthesia management, and immediate postoperative care. In contrast, in high-income countries, nurse anesthetists may work independently but frequently collaborate with specialized anesthesiologists. Their scope of practice in these settings may be limited by government regulations, institutional policies, or payment guidelines [33,34,35].

In Italy, among the anesthesiological procedures described in the international documents, Italian anesthesia nurses can autonomously manage the patient in the immediate postoperative period. However, other procedures, while directly performed, require a collaborative approach. This is reflected in the competency repertoire. Compared to European and international standards, the scope of autonomous practice for anesthesia nurses in Italy differs significantly. Due to the legal recognition of their responsibility for general nursing care (Law 739/94), the functions of Italian anesthesia nurses are not fully comparable to those of Certified Registered Nurse Anesthetists (CRNAs) in the United States or anesthesia nurses in France, the United Kingdom, or Switzerland. Nevertheless, the repertoire presented in this research enhances the international guidelines [33,34,35] by consolidating educational components (knowledge, abilities/skills, and learning outcomes) and practice standards (tasks/activities) into a single comprehensive document. This integration provides a detailed overview of competencies within specific activity areas.

### 4.4. Advancing Nursing Competency Frameworks in Operating Room Nursing: Final Considerations

Thanks to the comprehensive induction gained through at-home ethnography and multiple data collection methods [6,36], the research team successfully developed a comprehensive repertoire of technical professional specialist competencies for perioperative and perianesthesiological roles in operating room nursing. The competencies identified in this study included operating room nurse competencies previously described in an Italian research [8], while also expanding the number of competencies and distinguishing competencies between the two specific roles of operating room nurses: the perioperative nursing, where nurses govern all nursing interventions related to the surgical procedures and the instrumentation on the operating field, and the perianesthesiological nursing, where nurses govern all nursing interventions related to anesthesia conduct in the preparation room, operating room, and recovery room/post-anesthesia care unit. For the construction of this technical professional competency repertoire, the participants considered both the role related to specialist nursing care and the role related to advanced nursing care, as defined previously in the literature [37]. Following the comparison, it was decided to classify the identified technical professional competencies as specialist. This repertoire represents a unique contribution to nursing competency research and serves multiple purposes. It serves as a valuable tool for job descriptions and job analysis, useful for recruitment and placement, assessing and certifying specialized competencies for career development among operating room nurses, facilitating annual evaluations, and identifying areas for performance improvement in clinical practice. Additionally, it functions as a guiding framework for post-basic specialist academic training programs in operating room nursing education. It also helps individuals self-assess their competencies acquired through structured training programs or various learning experiences (work, self-training, internships, apprenticeships, etc.).

According to the EQF [6], recognition of qualifications for lifelong learning is feasible when qualifications are explicitly defined through specific criteria. The development of the proposed technical professional competencies repertoire marks a pivotal advancement toward achieving such recognition. The validation of competencies acquired through non-formal and informal settings remains challenging. This research addresses this challenge by contributing to the validation of non-formal and informal learning outcomes, thereby equipping the competent authority with a tool to facilitate the assessment of these outcomes against established standards for the two operating room nursing rules explored.

By clearly defining educational standards for training, the repertoire enhances the understanding of the complex practices in perioperative and perianesthesiological nursing. Furthermore, it supports the operationalization of processes for competency identification, validation, and certification across formal, non-formal, and informal learning contexts. This initiative is anticipated to guide the construction of subsequent competency repertoires, thereby enabling healthcare professionals to operate at elevated levels of practice and ensuring the recognition of their educational pathways, life experiences, and professional expertise throughout the work settings.

Future research should continue to complete the repertoire of competencies of operating room nursing by encompassing the integration of transversal competencies, the exploration of additional roles such as central sterilization nursing, and the development of an instrument to measure competency levels along with a checklist for monitoring task and activity completion for operating room nursing. The research team is currently developing a measurement instrument based on this study, aiming to incorporate proficiency classification levels of competency and autonomy akin to the privileges outlined by the Joint Commission International [38]. These privileges categorize qualifications into three levels: “autonomy; supervised autonomy; non-autonomy” and the research team is testing the validity of the measurement using this classification for each competency title, activity, knowledge, and skill detailed in the technical professional repertoire.

### 4.5. Limits

Among the limitations of this study, aside from the challenges associated with constructing the competency repertoire, the research team concentrated exclusively on identifying the technical professional specialist competencies pertinent to two specific operating room nursing roles. This focus precluded a thorough examination of competencies related to other roles, such as central sterilization nursing, as well as other essential competency areas, including transversal competencies, which are crucial for a holistic and comprehensive competency framework. Additionally, the involvement of participants was restricted to operating room nurses and masters of specialization nurses from a single hospital and one university, which may limit the generalizability of the findings. To mitigate this limitation, we included a diverse group of stakeholders—operating room nurses, nurse managers, and specialized nurses—to ensure a broader and more holistic perspective. While the study’s findings may be applicable to Italian operating room nurses based on the models of care used, they may not be fully generalizable internationally due to variations in regulatory frameworks.

## 5. Conclusions

The certification of competencies is essential for recognizing professional qualifications. There is significant variability in formal nursing education across Europe, particularly concerning specialist roles and competency development. Defining competencies acquired through informal and non-formal educational settings adds further complexity. This underscores the need to develop specific competency repertoires. This study addresses these challenges by proposing a repertoire of technical professional specialist competencies for perioperative and perianesthesiological nursing. Seventeen competencies were identified, each detailed with specific tasks/activities, knowledge, abilities/skills, learning outcomes, and areas of activity. Scholars, regulatory bodies, human resources professionals, nurse managers, and nurses can utilize this repertoire to establish and assess competency levels for the two operating room nursing roles studied. Future research should explore transversal competencies and professional competencies in additional operating room nursing roles.

## Figures and Tables

**Figure 1 healthcare-12-01774-f001:**
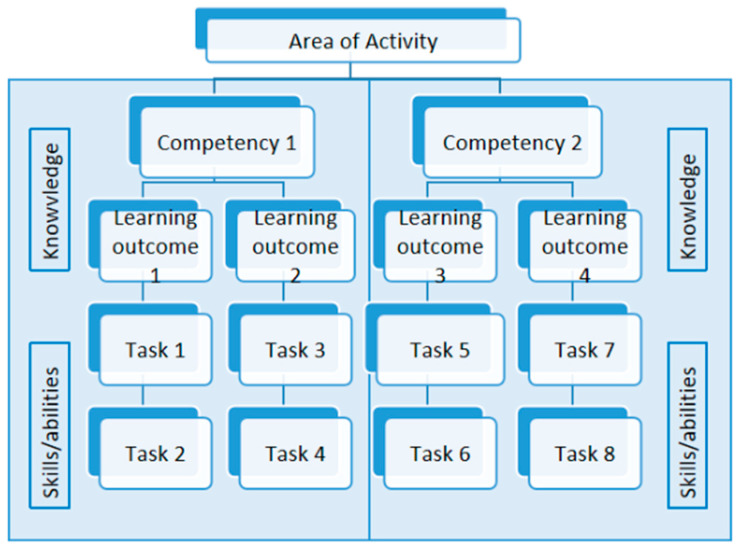
Fundamental elements that constitute a repertoire of competencies.

**Table 1 healthcare-12-01774-t001:** Sociodemographic sample characteristics (N = 46).

	N (%)
Sex	
Male	4 (8.7)
Female	42 (91.3)
Age (mean (SD))	36 (12.4)
Tenure in operating room (mean (SD))	12.68 (13.5) range (0–25)
Higher level of education	
Bachelor’s Degree	35 (76.1)
Specialization Master	10 (21.7)
Master’s Degree	-
Second level Specialization Master	1 (2.2)
Participating as	
Master’s student nurses	28 (60.8)
Nurses	12 (26.1)
Nurse managers	6 (13.1)

**Table 2 healthcare-12-01774-t002:** Summary of findings of the repertoire of technical and professional specialist competencies.

Area of Activity N = 6	Competencies N = 17	Learning Outcomes N = 19	Activities/Tasks N = 152	Abilities/Skills N = 218	Knowledge N = 222
1	4	4	28	49	64
2	2	3	16	20	23
3	2	2	15	20	25
4	2	3	26	30	23
5	2	2	14	18	22
6	5	5	53	81	65

Note: The numbers refer to the overall components identified in the repertoire for each thematic area. Refer to Appendix A for comprehensive details.

**Table 3 healthcare-12-01774-t003:** Summary of perioperative and perianaesthesiological competencies included in the repertoire.

Perioperative Nursing	Perianesthesiological Nursing
Preparing the perioperative environment Restoring the perioperative environment Admitting the assisted person in the perioperative and perianaesthesiological environment Transferring and positioning the assisted person in the perioperative and perianaesthesiological environment Preparing the assisted person for the surgical procedure Managing perioperative nursing care Promoting perioperative care continuity Documenting planned and delivered nursing care in the perioperative setting Restoring the perioperative environment Promoting professional development, competencies enhancement, and best practices in perioperative and perianaesthesiological settings	Preparing the perianesthesiological environment Restoring the perianaesthesiological environment Admitting the assisted person in the perioperative and perianaesthesiological environment Transferring and positioning the assisted person in the perioperative and perianaesthesiological environment Preparing the assisted person for anaesthesiological conduct Managing perianaesthesiological nursing care Promoting perianaesthesiological care continuity Documenting planned and delivered nursing care in the perianaesthesiological setting Promoting perianaesthesiological safety and risk management Promoting professional development, competencies enhancement, and best practices in perioperative and perianaesthesiological settings

**Table 4 healthcare-12-01774-t004:** Example of the complete repertoire of operating room competencies for one area of activity.

Area of activity 2: reception, transfer, and surgical positioning of the assisted person in the perioperative and perianesthesiological context	
Competency 5: Admitting the assisted person in the perioperative and perianesthesiological environment	
**Knowledge**	**Abilities/skills**
Characteristics and information contained in the operative note Methods and tools for the admission of the assisted person in the perioperative and perianaesthesiological context Nursing process of care, methods, and tools for identifying the care needs of the assisted person subjected to anesthesiological and/or surgical procedure Principles, processes, methods, and tools for taking in charge and formalizing the care agreement Behavior recommendations and best practices necessary for verification of the assisted person’s health and illness conditions, safety controls, and identification of risks attributable to the surgical procedure, instrumentation activities, and anesthesiological conduct activities Characteristics, structuring and information contained in general health documentation of hospitalization, nursing documentation, and specific documentation of the surgical procedure and anesthesiological conduct Behavior recommendations and best practices necessary to verify the presence and completeness of documentation Techniques, methods and tools for recording the assisted person’s entry into the perioperative and perianaesthesiological environment within the information systems Characteristics, structuring, and information contained in the sign-in section of the checklist Techniques, methods, and tools for completing the sign-in section of the checklist Techniques, methods, and tools for verbal, nonverbal, and paraverbal interaction and communication	Use the information contained in the operative note Use methods and tools for the admission of the assisted person in the perioperative and perianaesthesiological context Apply the nursing process of care and identify the care needs of the assisted person subjected to anesthesiological and surgical procedure Apply principles, processes, methods, and tools for the taking in charge and formalizing the care agreement Adopt behaviors in line with recommendations to verify the assisted person’s health and illness conditions, safety controls, and identify risks attributable to surgical procedure, instrumentation activities, and anesthesiological conduct activities Apply techniques to adhere to behavior recommendations and good practices to verify the presence and completeness of documentation Apply techniques and use methods and tools for recording assisted person admission in the perioperative and perianesthesiological context within information systems Apply techniques and use methods and tools to complete the sign-in portion of the checklist Use methods, tools, and effective interaction techniques Use methods, tools, and effective communication techniques to clearly convey information and receive it
Learning outcome 1: Effectively manages the admission process, demonstrate appropriate interpersonal relationship, and accurate documentation	
**Activities/tasks**	
Review the operative notes to gather information about the assisted person, the surgical procedure, and the anesthesiological conduct Receive the assisted person in the preparation room, operating room, and recovery room; greet him/her, introduce yourself, and explain the perioperative and perianesthesiological environment. Ensure the completeness of documentation for the surgical procedure, anesthesiological conduct, and the nursing care, including informed consent and instrumental and laboratory tests, gathering all necessary information also through observation and interviews Ensure that the assisted person’s health conditions allow proceeding with the planned surgical procedure and the agreed-upon anesthesiological conduct Record the admission of the assisted person (date and time) and update the necessary information/data each time, using the documentation systems available	
Learning outcome 2: Utilize a range of strategies (observation, dialogue, and interview) to manage and prioritize care needs and safety	
**Activities/tasks**	
Verify in the preparation room and in the operating room the patient identity, the surgical site, the procedure, and the agreed-upon anesthetic management. Check if the surgical site has been marked and if the preparation for surgery and anesthesia in the operating unit has been completed. Gather information regarding fasting, administration of pharmacological therapy, removal of jewelry and removable prostheses. In the recovery room, verify the identity of the assisted person, the type of surgical procedure performed, the type of anesthetic management adopted, the diagnostic procedures performed, the therapeutic procedures applied, and the positioning of medical devices. Take charge of the care needs of the assisted person in the preparation room, operating room, and recovery room, considering the information and data acquired from the operative notes and specific documentation regarding anesthetic management, surgical procedure, and related procedures performed and to be performed. Proceed with the planning of nursing care in all its phases: assessment, diagnoses, determination of outcomes, interventions, implementation, and evaluation of outcomes. Inform the patient about all nursing services that will be performed during the perioperative and perianesthetic period and formalize the care agreement, taking into account the expressed consent/disagreement, criteria of appropriateness, transparency, correctness, and compliance with current regulations on the protection of personal data. Work, cooperate, and collaborate with the anesthesiologist to verify that anesthesia safety checks have been completed. Operate, cooperate, and collaborate with the operating room team in identifying risks to the patient related to specific nursing activities involving instrumentation and anesthesia (difficulty managing airways and/or aspiration, infectious risk and/or risk of allergic reactions, risk of deep vein thrombosis (DVT), risk of postoperative nausea and vomiting (PONV), risk of pain onset, risk of blood loss, risk of physical and skin injuries (including corneal, pressure, friction, and slipping injuries), risk of anxiety onset, risk of thermal disturbances (hypothermia and hyperthermia), radiation risk, and electrical risk).. Complete the nursing area of the operating room checklist section (sign-in)	
Competency 6: Transferring and positioning the assisted person in the perioperative and perianesthesiological environment	
**Knowledge**	**Abilities/skills**
Characteristics of operating tables and accessory medical devices (antidecubitus, arm supports, shoulder supports, leg supports, belts, etc.). Principles of functionality and integrity of operating tables, accessory medical devices, and verification techniques. Characteristics and operation of the transport system. Principles, methods, and techniques of safe and functional transfer and transport of the assisted person in perioperative and perianesthesiological settings, Different types of postures/positions according to the surgical, anesthesiological, and nursing procedures to be performed. Behavior recommendations and best practices regarding surgical positioning of the person on the operating table. Principles and techniques of safe and functional surgical positioning of the assisted person on the operating table. Principles and methods of prevention of risks, accidental falls, trauma, and physical and skin injuries. Methods and tools for verifying the functional posture assumed by the person. Principles and methods for promoting person comfort and ensuring privacy during all transfers and positioning. Methods and techniques of multiprofessional cooperation. Techniques, methods, and tools for verbal, nonverbal, and paraverbal interaction and communication.	Use methods, tools, and techniques to verify the functionality and integrity of operating tables. Use the principles of functionality and integrity to proceed with the selection of operating table and ancillary medical devices. Use the transport system and apply the principles, behaviors recommended, and techniques of safe and functional transfer and transport of the assisted person in the perioperative and perianesthesiological setting. Apply recommended methods and techniques of safe and functional positioning of the assisted person on the operating table while operating, cooperating, and collaborating with the operating room team. Apply principles and methods prevention of risks, accidental falls, trauma, and physical and skin injuries. Use methods and techniques to verify the functional posture assumed by the person throughout the perioperative and perianesthesiological period. Apply methods to promote comfort and ensure the person’s privacy. Use multiprofessional cooperation techniques during transfers, transports, positioning, and verification of postures assumed by the person. Use methods, tools, and effective interaction techniques. Use methods, tools, and effective communication techniques to clearly convey information and receive it.
Learning outcome 3: Actively participates in the safe transfer of patients across the units and contributes to multiprofessional teamwork.	
**Activities/tasks**	
Work, cooperate, and collaborate with colleagues and the entire team in choosing the operating table on which to transfer the patient, according to the surgical procedure to be performed. Work, cooperate, and collaborate with the entire team in selecting the necessary medical devices (antidecubitus, arm supports, shoulder supports, leg supports, belts, etc.) based on the anesthetic management and surgical procedure, verifying the integrity and functionality, and optimizing the operating table according to the requirements. Work, cooperate, and collaborate with colleagues from the Units or Intensive Care Units, and the entire operating room team, during all transfers of the patients from the bed to the transport system, from the transport system to the operating table, from the operating table to the transport system, and from the transport system to the ward bed throughout the perioperative and perianesthesiological period. This includes when the person arrives from the Ward Unit or Intensive Care Unit to the operating block and is transported to the preparation room, the designated operating room, and the recovery room, and subsequently discharged back to the Ward Unit or Intensive Care Unit, ensuring the functionality of all necessary medical devices as required. Work, cooperate, and collaborate with the entire team in positioning the assisted person on the operating table, considering the different nursing procedures, anesthesiological procedures, and surgical procedures required during the entire perioperative and perianesthesiological period. Ensure the appropriate positioning of the patient on the operating table and reduce that it interferes with vital functions. Verify the proper functionality of the monitoring system and periodically check vital signs, common clinical signs and symptoms through physical examination (as appropriate), and positions assumed throughout the perioperative and perianesthesiological period. Prevent risks such as accidental falls, trauma, and physical and skin injuries (from pressure, friction, or stretching/slipping). Promote the comfort of the person and ensure privacy during all transfers and positioning, always considering the patient’s level of mobility and joint freedom, as well as the level of understanding and cooperation.	

Notes: This area of activity encompasses two distinct competencies. Each competency is characterized by specific knowledge and skills directly associated with it. The knowledge and skills are connected to specific learning outcomes. The achievement of learning outcomes is measured through specific tasks and activities.

## Data Availability

The data presented in this study are available on request from the corresponding author due to privacy restrictions.

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
