# Peer review of "Toward the Definition of a Repertoire of Technical Professional Specialist Competencies for Operating Room Nurses: An Ethnographic Study"

_healthcare, 2024, doi:10.3390/healthcare12171774_

Round 1
Reviewer 1 Report
Comments and Suggestions for Authors
This is a well-written piece of work.
For further clarity, table 3 (summary of competencies included in the repertoire) could be separated e.g. 1 -10 for each role to reflect the discussion on the previous page. Shared competencies will have the same number for both roles and the content will remain the same.
Also, the learning outcomes could be framed to state the knowledge and skills to be acquired in a concise way without repeating the contents of the other columns, for example:
LO 1 Effectively manages the admission process, demonstrate appropriate interpersonal relationship, and accurate documentation.
LO 2 - Utilise a range of strategies (observation, dialogue, and interview) to manage and prioritise care needs and safety.
Lo 3 - Actively participates in the safe transfer of patients across the units and contribute to multiprofessional teamwork.
Comments on the Quality of English LanguageThe quality of English is good.
Author Response
Reviewer 1
Answer: Thank you for dedicating your time to our manuscript and for the precious suggestions.
For further clarity, table 3 (summary of competencies included in the repertoire) could be separated e.g. 1 -10 for each role to reflect the discussion on the previous page. Shared competencies will have the same number for both roles and the content will remain the same.
Answer: Thank you for the suggestion. We specified the competencies for each role, creating two columns in the table to enhance clarity as suggested.
Also, the learning outcomes could be framed to state the knowledge and skills to be acquired in a concise way without repeating the contents of the other columns, for example:
LO 1 Effectively manages the admission process, demonstrate appropriate interpersonal relationship, and accurate documentation.
LO 2 - Utilise a range of strategies (observation, dialogue, and interview) to manage and prioritise care needs and safety.
Lo 3 - Actively participates in the safe transfer of patients across the units and contribute to multiprofessional teamwork.
Answer: Thank you for the suggestion, we adapted the reduced version of the learning outcomes as suggested in Table 4.
Reviewer 2 Report
Comments and Suggestions for Authors
My compliments to the authors for undertaking this audacious manuscript and researching the operating room nursing competencies, let alone all the activities, skills, etc., for more than basic nursing in the OR. Is it possible to separate the peri-operative roles from the anesthesia CRNA roles? Additionally, PACUs have their own professional guidelines that might be templates for European operating room nurses. The US also has Microsoft™ Excel spreadsheets as templates for various nursing positions that might be useful (references).
What type of documentation form was used for the At-Home Ethnographical data collection?
Are the study results applicable to generalization?
Table 4: Supplementary Table represented more than one activity. Line 438 review / revise if indicated.
* Originality / Novelty: This study is not a novelty. It has been identified as necessary to update one aspect of the nursing profession: the Operating room. The authors provide a compelling argument for this data collection, analysis, and manuscript. The reading audience will encompass more than the per-operative profession.
* Significance of Content: the discussion involving the numerical results of the study is representative of the extensive work involved in this study. The competencies identified, along with the skills, tasks, knowledge, etc., are relevant to the profession as a whole.
*Quality of Presentation: While this presentation goes into detail and depth, reading the manuscript in one sitting contains much to ponder and digest. The authors spare very little detail in their presentation. Kudos.
* Interest to the readers: As stated, this manuscript has a definite reading audience, and those in the perioperative and other areas identified will find the details useful for their professions.
Comments on the Quality of English LanguageVery little was identified. Spell checking and a few grammar-sentence structures.
Author Response
Reviewer 2
My compliments to the authors for undertaking this audacious manuscript and researching the operating room nursing competencies, let alone all the activities, skills, etc., for more than basic nursing in the OR.
Answer: Thank you for dedicating your time to our manuscript and for the precious suggestions.
Is it possible to separate the peri-operative roles from the anesthesia CRNA roles? Additionally, PACUs have their own professional guidelines that might be templates for European operating room nurses. The US also has Microsoft™ Excel spreadsheets as templates for various nursing positions that might be useful (references).
Answer: Thank you for your suggestion. The peri-anesthesiological operating room nursing competencies identified in our research do indeed share similarities with the roles of Certified Registered Nurse Anesthetists (CRNAs) and those working in Post-Anesthesia Care Units (PACUs). However, it is important to note that in Italy, these specializations within nursing are not formally recognized, and their scope of practice is more limited compared to the specialized roles you mentioned. We have now incorporated this aspect into the discussion section.
What type of documentation form was used for the At-Home Ethnographical data collection?
Answer: Thank you for your observation. We integrated this information in the data collection session.
Are the study results applicable to generalization?
Answer: Thank you for your observation. We integrated information regarded in the generalizability of findings in the limits session.
Table 4: Supplementary Table represented more than one activity. Line 438 review / revise if indicated.
Answer: Thank you for your observation. We have revised Table 4 to enhance clarity and, following these revisions, it has been renamed Table 5. We hope that the updated table is now more easily understandable for the reader.
* Originality / Novelty: This study is not a novelty. It has been identified as necessary to update one aspect of the nursing profession: the Operating room. The authors provide a compelling argument for this data collection, analysis, and manuscript. The reading audience will encompass more than the per-operative profession.
* Significance of Content: the discussion involving the numerical results of the study is representative of the extensive work involved in this study. The competencies identified, along with the skills, tasks, knowledge, etc., are relevant to the profession as a whole.
*Quality of Presentation: While this presentation goes into detail and depth, reading the manuscript in one sitting contains much to ponder and digest. The authors spare very little detail in their presentation. Kudos.
* Interest to the readers: As stated, this manuscript has a definite reading audience, and those in the perioperative and other areas identified will find the details useful for their professions.
Answer: Thank you very much. We are grateful to your suggestions for improvement.
Reviewer 3 Report
Comments and Suggestions for Authors
ABSTRACT: Include the year in which the data was collected.
ABSTRACT: include form of data analysis.
INTRODUCTION: I suggest that the introduction be summarized. It is long and takes up too much time. I suggest that the authors find a shorter way of presenting the different competence systems.
INTRODUCTION: For the authors to be able to draw up a study that addresses the competences expected in nursing in a single hospital in a city in Italy, it doesn't seem necessary to discuss in the introduction the competences expected in nursing in the perioperative environment on the entire European continent. It would seem a disproportionate discussion. It seems more coherent to talk about the competences expected in Italy, without the need for major comparisons with other European countries, since other countries may even have their own different regulations for nursing.
INTRODUCTION: include the concept of technical competence and professional competence.
METHODS: Exclusion criteria are not the opposite of inclusion criteria. You have to think about which characteristics would be exclusionary among those who would be suitable for inclusion.
METHODS: What training do the people responsible for data collection have?
TABLE 2: I suggest adding a legend identifying the areas of activity in the first column.
METHODS: conceptualize "knowledge area". conceptualize "skill".
TABLE 4 AND SUPPLEMENTARY FILE: I suggest that the authors edit the tables to present a didactic look that makes it easier to understand the divisions of the competencies. As they are long tables, these edits help the reader not to get confused while reading.
DISCUSSION: I suggest that the authors include in the discussion what the council that regulates the nursing profession in Italy recognizes as the attributions of nursing in the operating room in perioperative and perianesthesiological functions. This will give an idea of the scope of the legal provisions for nursing in this environment. It is an important anchor for consolidating the attributions raised by the participants. Each country can define different attributions and autonomies for nursing.
DISCUSSION: Is the comparison made of specific perioperative competences appropriate for undergraduate courses or are they expected to be developed in postgraduate courses? In this discussion, it is important to consider the priorities of the competencies used throughout undergraduate nursing courses to train generalist nurses.
DISCUSSION: I suggest that the authors discuss less about the potential of the methodology used. They should explore the results more, comparing the differences and novelties more clearly with the other repertoires of competences.
LIMITATIONS: there is no need to list the strengths of the research, these are already described throughout the study. It gets repetitive.
Author Response
Reviewer 3
Answer: Thank you for dedicating your time to our manuscript and for the precious suggestions.
ABSTRACT: Include the year in which the data was collected. Answer: Thank you for your observation. We integrated this information in the abstract.
ABSTRACT: include form of data analysis. Answer: Thank you for your observation. We integrated this information in the abstract.
INTRODUCTION: I suggest that the introduction be summarized. It is long and takes up too much time. I suggest that the authors find a shorter way of presenting the different competence systems.
Answer: Thank you for your observation. We have reviewed the Introduction section and have worked to synthesize and streamline the concepts presented.
INTRODUCTION: For the authors to be able to draw up a study that addresses the competences expected in nursing in a single hospital in a city in Italy, it doesn't seem necessary to discuss in the introduction the competences expected in nursing in the perioperative environment on the entire European continent. It would seem a disproportionate discussion. It seems more coherent to talk about the competences expected in Italy, without the need for major comparisons with other European countries, since other countries may even have their own different regulations for nursing.
Answer: Thank you for your observation. We have reviewed the Introduction section and expanded the description of operating room nursing in Italy. Considering that documentation on the competencies of operating room nurses in Italy is limited, we referred to international professional societies to describe the currently recognized areas of competence. Additionally, since the specialization relates to post-basic education, it was necessary to introduce the educational system and certification of competencies in both Europe and Italy. These regulations guide the certification and recognition process and serve as the framework for our study.
INTRODUCTION: include the concept of technical competence and professional competence.
Answer: Thank you for your suggestion. We have integrated the definitions of competency, as well as technical professional competencies, into the Methods section under the Theoretical Framework subheading.
METHODS: Exclusion criteria are not the opposite of inclusion criteria. You have to think about which characteristics would be exclusionary among those who would be suitable for inclusion.
Answer: Thank you for your observation. We specified better the inclusion and exclusion criteria.
METHODS: What training do the people responsible for data collection have?
Answer: Thank you for your observation. We integrated this information in the methods section
TABLE 2: I suggest adding a legend identifying the areas of activity in the first column.
Answer: Thank you for your suggestion. The areas of activity are described in detail in lengthy sentences, and including this information directly in the notes would make the table legend excessively long. Instead, we have added a note directing readers to refer to Supplementary Table 3 for comprehensive details.
METHODS: conceptualize "knowledge area". conceptualize "skill".
Answer: Thank you for your observation. We integrated all the definitions of terms used in the repertoire in the methods section.
TABLE 4 AND SUPPLEMENTARY FILE: I suggest that the authors edit the tables to present a didactic look that makes it easier to understand the divisions of the competencies. As they are long tables, these edits help the reader not to get confused while reading.
Answer: Thank you for your observation. We have revised Table 4 to enhance clarity and, following these revisions, it has been renamed Table 5. We hope that the updated table is now more easily understandable for the reader.
DISCUSSION: I suggest that the authors include in the discussion what the council that regulates the nursing profession in Italy recognizes as the attributions of nursing in the operating room in perioperative and perianesthesiological functions. This will give an idea of the scope of the legal provisions for nursing in this environment. It is an important anchor for consolidating the attributions raised by the participants. Each country can define different attributions and autonomies for nursing.
Answer: Thank you for your valuable observation. We have added information about the laws and regulations governing the nursing profession in Italy, highlighting that these laws apply generally to nursing as a whole. Currently, there is no specific legal recognition or defined specialization for operating room nursing in the perioperative and perianaesthesiological contexts.
DISCUSSION: Is the comparison made of specific perioperative competences appropriate for undergraduate courses or are they expected to be developed in postgraduate courses? In this discussion, it is important to consider the priorities of the competencies used throughout undergraduate nursing courses to train generalist nurses.
Answer: Thank you for your insightful observation. In Italy, the development of technical professional competencies for operating room nursing typically occurs during post-basic education. Currently, operating room nurses enter their positions with generalist nursing training and receive additional on-the-job training. Specialization in this field is not a formal requirement for job placement. This situation raises important questions about which competencies should be prioritized in undergraduate nursing programs aimed at training generalist nurses. While some training is provided in areas like sterile field management, anesthesia medications, and post-surgery care, it is not feasible to cover the full spectrum of specialized competencies required for operating room nursing within a bachelor's degree. To ensure safe and high-quality care, there is a need to legally recognize and support the specialization and post-basic education of nurses. Relying solely on generalist nurses, expected to handle every aspect of care, may not be sufficient. Specialized experts are essential in this complex and demanding field.
DISCUSSION: I suggest that the authors discuss less about the potential of the methodology used. They should explore the results more, comparing the differences and novelties more clearly with the other repertoires of competences.
Answer: Thank you for your valuable observation. We have revised the discussion to include more in-depth comparisons with other operating room nursing competency frameworks, highlighting the differences and novelties in our findings.
LIMITATIONS: there is no need to list the strengths of the research, these are already described throughout the study. It gets repetitive.
Answer: Thank you for your observation. we removed strengths.
Reviewer 4 Report
Comments and Suggestions for Authors
1. Issue very interesting for the reader
2. As far as Introduction section is concerned, even though it is really consistent, however I think it is too extensive.
3. As regard the methodology section, I would like to mention some aspects:
a. Selection criteria.
* The authors should explain in more detail the reasons why" those participants who have worked for at least 2 years" is an inclusion criterion. And why " having less than 2 years of experience" and "less than 500 hours of internship...." are considered exclusion criteria. Providing this information to the reader is essential in qualitative studies.
* How was the recruitment process? I mean, How was the approach?. This is not described in the manuscript.
* How many people refused to participate or dropped out ? Reasons?
b. Data collection
* As regard data collection phase three, according to the autors, 34 interviews were carried out in the study. However, the tool used to collect information have not been shown in supplementary material.
* How many discussion groups were carried out in this research?
* In which phase of the study were the discussion groups conducted?
*How many participants made up each discussion group?
*what was the duration of discussion groups?.
*In addition to this, the questions guide from which the authors obtained information have not been included in the manuscript or supplementary material.
4. Regarding the results section,
* Despite the fact that the authors explain in detail some aspects of data analysis ( lines: 343-352), however, the codes from the participants´discources are missing. Consequently, I am not aware of how the themes and subthemes have been identified from the data. I honestly think that this aspect would contribute to verifying the rigour and consistence of this research.
* If the authors do not present the participant quotations to illustrate the themes and subthemes that come out from the data analysis, It could be interpreted as suggesting that this research has been based solely on participant observation and meetings to triangulate the information with the research team.
* Table 4 is difficult to be interpreted properly . Its presentation should be improved. For instance: Activities/ Tasks/ Knowledge/Abilities /Skills are not well aligned and consequently, the information given is confusing and some details are lost.
5. As for the discussion section,
* In this section, what I observe is a description of the results. The comparison with the existing literature is scarce. This make the section weaker.
6. As regard ethical consideration,
* The Consent Informed is not shown in supplementary material,
* The protocol code is not included in the Institutional Review Board Statement
Comments on the Quality of English Language
* Some sentences of the manuscript are too long. For instance: from line 546 to 550.
Author Response
Reviewer 4
- Issue very interesting for the reader
Answer: Thank you for dedicating your time to our manuscript and for the precious suggestions.
- As far as Introduction section is concerned,even though it is really consistent, however I think it is too extensive.
Answer: Thank you for your observation. We have reviewed the Introduction section and have worked to synthesize and streamline the concepts presented.
- As regard the methodology section, I would like to mention some aspects:
- Selection criteria.
* The authors should explain in more detail the reasons why" those participants who have worked for at least 2 years" is an inclusion criterion.
Answer: Thank you for your observation. We integrated this information in the methods section.
And why " having less than 2 years of experience" and "less than 500 hours of internship...." are considered exclusion criteria. Providing this information to the reader is essential in qualitative studies.
Answer: Thank you for your observation. We integrated this information in the methods section.
* How was the recruitment process? I mean, How was the approach?. This is not described in the manuscript.
Answer: Thank you for your observation. We integrated this information in the methods section.
* How many people refused to participate or dropped out ? Reasons?
Answer: Thank you for your observation. We integrated this information in the methods section.
- Data collection
* As regard data collection phase three, according to the autors, 34 interviews were carried out in the study. However, the tool used to collect information have not been shown in supplementary material.
Answer: Thank you for your observation. We have added a supplementary file Table 2 containing the open-ended questions used in the interview to the double and Table 1 with the questions posed during group discussions. Since both data collection methods involved open-ended questions and unstructured interviews, additional questions were asked based on participants' responses. These supplementary materials should now provide a clear overview of the data collection tools used in the study.
* How many discussion groups were carried out in this research?
Answer: Thank you for your observation. We integrated the information in the methods section.
* In which phase of the study were the discussion groups conducted?
Answer: Thank you for your observation. We specified in the methods section that the group discussions were part of phase two of the study.
*How many participants made up each discussion group?
Answer: Thank you for your observation. We specified this information in the methods section.
*what was the duration of discussion groups?.
Answer: Thank you for your observation. We specified this information in the methods section.
*In addition to this, the questions guide from which the authors obtained information have not been included in the manuscript or supplementary material.
Answer: Thank you for your observation. We have clarified in the methods section that the open-ended questions addressed during the focus group discussions are provided in Supplementary Table 1.
- 4. Regarding the results section,
* Despite the fact that the authors explain in detail some aspects of data analysis ( lines: 343-352), however, the codes from the participants´discources are missing. Consequently, I am not aware of how the themes and subthemes have been identified from the data. I honestly think that this aspect would contribute to verifying the rigour and consistence of this research.
Answer: Thank you for your observation. To address your concern and enhance the transparency and rigor of our research, we have added Table 4 in the results section, which details the themes, subthemes, and corresponding codes derived from the participants' discourses.
* If the authors do not present the participant quotations to illustrate the themes and subthemes that come out from the data analysis, It could be interpreted as suggesting that this research has been based solely on participant observation and meetings to triangulate the information with the research team.
Answer: Thank you for your observation. We have revised the results section to include key participant quotations that illustrate the main themes and subthemes, providing clearer examples of the "interview to the double" method. This addition enhances transparency and clarity regarding how the competency repertoire was generated.
* Table 4 is difficult to be interpreted properly . Its presentation should be improved. For instance: Activities/ Tasks/ Knowledge/Abilities /Skills are not well aligned and consequently, the information given is confusing and some details are lost.
Answer: Thank you for your observation. We have revised Table 4 to enhance clarity and, following these revisions, it has been renamed Table 5. We hope that the updated table is now more easily understandable for the reader.
- As for the discussion section,
* In this section, what I observe is a description of the results. The comparison with the existing literature is scarce. This make the section weaker.
Answer: Thank you for your observation. We have strengthened the discussion section by incorporating additional references and comparisons with existing competency documents. However, it's important to note that there is limited peer-reviewed research specifically on competency development in this field. Most available studies focus on the development of instruments that only partially measure competency areas, without a deep exploration of the tasks, activities, or the knowledge and skills required to fully mobilize these competencies. Therefore, while we conducted a thorough literature review, we found limited opportunities for comparison with scientific research. Instead, we have compared our findings with operating room nursing association guidelines and official documents.
- 6. As regard ethical consideration,
* The Consent Informed is not shown in supplementary material,
Answer: Thank you for your observation. We sent the consent form to the editorial team.
* The protocol code is not included in the Institutional Review Board Statement
Answer: Thank you for your observation. The approval from the university and the board of directors of the institution did not have a specific number to report. We shared the document with the editorial team.
Comments on the Quality of English Language
* Some sentences of the manuscript are too long. For instance: from line 546 to 550.
Answer: Thank you for your observation. We checked and modified the language whenever possible.
Round 2
Reviewer 3 Report
Comments and Suggestions for Authors
Congratulations to the authors, there have been important changes to the manuscript. The additions they made were very pertinent. The legal provisions included in the text are essential to legitimize the activities carried out by nursing in this surgical environment. The study provides a careful portrait of the reality of nursing in the surgical center, and they carefully employed the methodological sense of ethnography.
However, the text of the introduction and discussion are long. I suggest that the authors create subtopics to divide the main ideas of the introduction and discussion. This will help to organize and delimit the central ideas of the manuscript. Reflect on the main pillars of the results and how you want to highlight them in the discussion. The use of subtopics can help to provide greater clarity on the central ideas discussed in the manuscript.
Author Response
Congratulations to the authors, there have been important changes to the manuscript. The additions they made were very pertinent. The legal provisions included in the text are essential to legitimize the activities carried out by nursing in this surgical environment. The study provides a careful portrait of the reality of nursing in the surgical center, and they carefully employed the methodological sense of ethnography.
Answer: Thank you for dedicating your time to our manuscript and for the precious suggestions.
However, the text of the introduction and discussion are long. I suggest that the authors create subtopics to divide the main ideas of the introduction and discussion. This will help to organize and delimit the central ideas of the manuscript. Reflect on the main pillars of the results and how you want to highlight them in the discussion. The use of subtopics can help to provide greater clarity on the central ideas discussed in the manuscript.
Answer: Thank you for your valuable suggestion. After careful consideration, we recognized the importance of providing a clear and structured presentation of the key concepts in both the introduction and discussion sections. To enhance clarity and better organize the content, we have revised the manuscript by introducing subtopics in the introduction. These subtopics allow us to effectively break down the information related to the phenomenon from national, international, and theoretical perspectives.
Similarly, in the discussion section, we have reflected on the main pillars of our results and structured the discussion in subtopics to delineate and highlight the key points and to improve the overall readability and coherence of the manuscript.
We believe these changes will make the manuscript more accessible and easier to navigate for readers.
Reviewer 4 Report
Comments and Suggestions for Authors
As for results section, in my opinion table 4 should be included in suplementary material. Its inclusion in the main text would make the results section overly extensive.
As regard table 5, as I said in the previous review, it is still a bit confusing to be interpreted. Please, try to improve it.
As far as ethical considerations are concerned, I still miss the protocol code of this research. In my opinion, all research studies evaluated by an ethics committee must have a protocol code.
Author Response
Answer: Thank you for dedicating your time to our manuscript and for the precious suggestions.
As for results section, in my opinion table 4 should be included in suplementary material. Its inclusion in the main text would make the results section overly extensive.
Answer: Thank you for the suggestion. We moved Table 4 to supplementary material.
As regard table 5, as I said in the previous review, it is still a bit confusing to be interpreted. Please, try to improve it.
Answer: Thank you for the suggestion. We re-propose a new version of the table that now is named Table 4. We added a note to the table to better explain the reading. We are confident that now it should result easier to the reader.
As far as ethical considerations are concerned, I still miss the protocol code of this research. In my opinion, all research studies evaluated by an ethics committee must have a protocol code.
Answer: Thank you for the suggestion. We integrated the protocol number of the approval.